# Dynamic Shear Texture Evolution during the Symmetric and Differential Speed Rolling of Al-Si-Mg Alloys Fabricated by Twin Roll Casting

**DOI:** 10.3390/ma17010179

**Published:** 2023-12-28

**Authors:** Jae-Hyung Cho, Geon-Young Lee, Seong-Ho Lee

**Affiliations:** 1Korea Institute of Materials Science, 797 Changwondaero, Seongsan-gu, Changwon 51508, Republic of Korea; arebee@kims.re.kr (G.-Y.L.); lsh960608@kims.re.kr (S.-H.L.); 2School of Mechanical Engineering, Pusan National University, 2 Busandaehak-ro 63beon-gil, Geumjeong-gu, Busan 46241, Republic of Korea

**Keywords:** Al-Si-Mg aluminum alloys, differential speed rolling, reduction in area (RA), crystallographic textures, crystal plasticity, finite element method

## Abstract

The effects of a reduction in area (RA) and the speed ratio between the top and bottom rolls on a shear strain and the crystallographic texture evolution of Al-Si-Mg (1.0%Si-0.6%Mg) aluminum alloys fabricated by twin roll casting (TRC) were comprehensively examined experimentally and through numerical predictions. Initial twin-roll casted strips had a texture gradient from the surface to the center. 〈111〉//ND textures were found in the surface layer, and weak rolling textures existed in the center of the strip. The distributions of shear and plane strain compression (PSC) textures varied with the RA and differential speed ratio. Strong shear textures including a rotated cube, {100}〈011〉, were obtained from both the symmetric and differential speed rolling processes. Symmetric rolling with a different reduction in area (SRDRA) produced shear textures mainly in the surface layer. Differential speed rolling (DSR) caused dynamic shear textures along the thickness direction, not limited to the surface. Based on the finite element method and crystal plasticity, the effects of different RA values, differential speed ratios, and friction coefficients on shear strain and texture evolution are discussed in detail. Loads measured from work rolls during DSR decreased with an increase in the speed ratio.

## 1. Introduction

Rolling is a conventional process that is used to fabricate sheet products from many wrought metals, including Al, Mg, Fe, and Ti alloys. Conventional rolling usually contains both plane strain compression (PSC) and shear deformation [1]. Although the rolling process is assumed to be similar to PSC, shear deformation due to the roll gap geometry and friction between the rolls and materials is unavoidable [2]. The center region of the rolled sheets approaches PSC, and the surfaces of the sheets experience more shearing. Differences in the deformation mechanisms between the center and surface regions frequently result in texture gradients or inhomogeneities [3].

The mechanical and electrical properties of textured sheets strongly depend on the crystallographic orientations. Improvement of the r-value of aluminum sheets has been a critical issue, and it has been shown that the shear texture introduced by shearing affects the mechanical responses [4,5,6,7,8]. The influence of a texture gradient across the thickness on the r-value profile of Al-Mg-Si alloy sheets has been studied, and Goss orientation was found to have the maximum impact on the r-value profiles [9]. The preferred orientations of austenite and ferrite phases in duplex steel are different from those of single-phase austenitic and ferritic steels due to mutual interaction between them [10]. To improve the performance of 3.0% Si-oriented silicon steel, the effects of cold-rolling deformation and the rare earth element Y on the microstructure and texture have also been analyzed [11]. Cold-rolled sheets containing the rare earth element Y had greater shear bands than Y-free steels, and this resulted in more γ texture strengthening, which was effective for Goss nuclei. The microstructure and texture evolution of aluminum plates produced by multi-pass cold rolling and graded annealing were also investigated [12]. The texture of the cold-drawn copper wire was investigated using electron backscatter diffraction. There is some similarity in crystallographic textures between wire and rolled sheets. The fiber texture of the central region of the wire was considered as the rolling textures, and that of the periphery region was revealed to be similar to the shear texture [13].

An asymmetric rolling (ASR) process, such as a process with different roll diameter rolling (DDR) or differential speed rolling (DSR) characteristics, effectively gives rise to shear deformation of the rolled strips. Using rolling mills with different roll diameters, the evolution of the deformation texture was examined and compared with model predictions [14]. The ideal shear texture increased with an increase in the ratio between the shear strain and normal strain. The shear textures of a rotated cube, {001}〈011〉, and 〈111〉//ND fiber developed during the DDR of AA1050 remained almost unchanged when annealed at 400 °C for 1 h [15]. Intensity weakening or randomization of the shear textures during the annealing of AA6xxx was also observed [16,17]. To gain a better understanding of the enhanced formability of the Al-Mg-Si alloy used for automotive outer panels, the shear texture evolution of twin-roll casted AA6016 sheets during ASR was investigated, with the results showing that the formability of ASR sheets was significantly improved [18]. Effects of the differential roll speeds on the microstructure, texture, and mechanical properties of 7075 aluminum plates fabricated by a continuous casting and rolling (CCCR) process were investigated [19]. It was found that the greater the difference in the roll speeds was, the greater the isotropy and the hardness of the final product were, although elongation minorly decreased. Asymmetric rolling of aluminum alloy AA6061 (Al-0.6 Si-1.0 Mg) was studied to investigate the effects of the core processing parameters on mechanical properties, and it was found that ASRed specimens exhibited superior tensile properties to symmetrically rolled counterparts [20]. The experimental description of asymmetric rolling applied to aluminum alloys and steels was also comprehensively reviewed [21]. Several aspects including process methods (i.e., the dissimilarity of the roll diameters, the rolls’ angular speed, or the friction conditions), the process parameters (i.e., the total thickness reduction, the thickness reduction per pass, the peripheral speed ratio, and the rolling routes) and their effect on the texture and microstructure evolution, and mechanical properties were addressed.

Various features of the ASR process were examined using experiments and model analyses. The bending curvature of the rolled strips varied with the roll parameters, specifically the speed ratio, reduction in area, and amount of friction [22,23,24,25,26]. Both positive and negative curvatures depended on the working parameters [27]. The bending curvatures of the strips were utilized to estimate the coefficient of friction [28]. Motion of the neutral points was evident in DSR, and the cross-shear zone varied in DSR [29]. Shear texture evolution was studied using aluminum single crystals with a cube texture component under high pressure torsion [30]. Using the finite element method (FEM) with a zooming analysis, an efficient elastic–plastic analysis to predict the metal plate texture after the cold rolling of a S45C plate was established [31].

Studies on the asymmetric rolling of aluminium strips fabricated by twin-roll casting was relatively few, compared to those fabricated by conventional ingot casting. In this study, Al-Si-Mg strips fabricated by twin-roll casting were cold-rolled using rolling mills with the same roll diameter for a comprehensive investigation of the effects of the rolling parameters of the reduction in area (RA) and the differential speed ratio on the shear distribution and crystallographic texture evolution. The RA of each rolling pass depends on the roll gap geometry, and this affects the shear deformation and texturing through the thickness direction of the sheets. The differential speed ratio between the top and bottom rolls also produces shear deformation. Based on the two major mechanisms that cause shear deformation during rolling, the effects of symmetric rolling with a different reduction in area (SRDRA) and differential speed rolling (DSR) characteristics on shear texturing were examined in detail. Finite element modeling of the rolling processes was carried out, and the predicted textures based on crystal plasticity were compared with the experimental results. In addition, the effects of friction between the rolls and sheets on the shear distribution and texturing were investigated.

## 2. Experimental Procedure

Al-Si-Mg (1.0%Si-0.6%Mg) aluminum strips were fabricated by twin-roll casting with an initial thickness of 4 mm. Various cold-rolling processes were carried out using a differential speed rolling (DSR) mill with a roll diameter of 280 mm. Two different experimental processes, symmetric rolling with a different reduction in area (SRDRA) and DSR, were carried out in order to investigate the texture evolution during the rolling processes. No lubrication was applied.

When carrying out the SRDRA process, the speeds of the top and the bottom rolls were identical. Five different rolling schedules (or different rolling passes) were applied to achieve a total RA of 75%: 17 passes at an approximately 8.0% RA for each rolling pass, 11 passes at an approximately 11.4% RA, 9 passes at an approximately 15.5% RA, 5 passes at an approximately 24.5% RA, and 3 passes at an approximately 36.0% RA. Working parameters of SRDRA were summarized in Table 1.

For the DSR process, the speed of the top roll was held constant at 5 mpm (meters per minute). The speed of the bottom roll varied from 5 (0.595 rad/s) to 10.0 mpm (1.19 rad/s). Therefore, the speed ratios between the top and bottom rolls varied from 1.0 to 2.0. After five passes, the final thickness of the sheet was approximately 1 mm, and the total reduction in area (RA) was 75%. The RA for each rolling pass was approximately 24.5%. Working parameters of DSR were summarized in Table 2.

Macroscopic texture measurements were carried out using XRD on the top, middle, and center regions of the rolled sheets. The top surface of the sheet here refers to the surface side contacting the top roll. The XRD system used was an X’pert Pro (PANalytical) with Cu-Kα radiation operating at 40 kV and 30 mA. Three incomplete pole figures of (111), (200), and (220) were collected on a $5^{{}^{\circ}}$ grid up to a sample tilt of $70^{{}^{\circ}}$. The obtained data were analyzed using the WIMV (William–Imhof–Matthies–Vinel) method with automated conditional ghost correction [32,33], assuming cubic crystal and triclinic sample symmetries.

## 3. Simulation Procedure

### 3.1. Modeling Rolling Process

Two different rolling processes, SRDRA and DSR, were modeled using two-dimensional FEM codes, DEFORM2D. During the SRDRA process, the top and bottom rolls operated at the same speed, and only half of the workpiece was modeled considering a symmetric plane. Under symmetric rolling conditions, the effects of various RAs on the shear deformation and texture evolution were predicted. Figure 1a presents a schematic symmetric domain with five elements through the thickness direction. To minimize the end effect of the model domain, the middle region of the sheet specified with the circle was focused on to investigate the texture evolution and strain distribution. Deformation gradients were extracted with time along six nodal points, and the mean values between the adjacent tracking material points were used to compute the velocity gradient. Considering half of the workpiece, each element is approximately 0.4 mm thick. In addition, under the working parameters, the workpieces quickly passed by the roll gap less than 0.2 s. The rolls were considered as a rigid body, and constant shear friction between the rolls and workpieces was assumed.

As the DSR process has different speeds for the top and bottom rolls, the whole thickness of the sheet was modeled with 10 elements through the thickness direction, as shown in Figure 1b. Along the 11 nodal points used, deformation gradients were extracted over time, similar to the symmetric model. The velocity gradients were computed using the typical relationship,
(1)$$L=\dot{F}\cdot F^{-1},$$
where L is the velocity gradient, and F is the deformation gradient.

### 3.2. Self-Consistent Polycrystal Model

Variations in mechanical properties are closely related to slip deformation in FCC (face-centered cubic) metals. Here, a viscoplastic self-consistent model (VPSC) [34] was applied to Al-Si-Mg alloys to understand the crystallographic texture evolution during the rolling process. The visco-plastic constitutive behavior in a single crystal is described with the non-linear rate-sensitivity equation below [35].
(2)$$\dot{\varepsilon_{ij}}=\frac{1}{2}\sum_{s=1}^{ns}\left(m_{ij}^{s}+m_{ji}^{s}\right)\;{\dot{\gamma }}^{s}=\frac{{\dot{\gamma }}_{0}}{2}\frac{1}{{\left(\tau_{0}\right)}^{1/m}}\sum_{s=1}^{ns}\left(m_{ij}^{s}+m_{ji}^{s}\right)\left(\sigma_{kl}\;m_{kl}^{s}\right){\left|\sigma_{pq}\;m_{pq}^{s}\right|}^{\frac{1}{m}-1}.$$Here, ${\dot{\gamma }}^{s}$ is the shear rate in the slip system (or variants) indexed by *s*. *ns* is the total number of operating slip systems. $m_{ij}^{s}$ is the symmetric part of the Schmid tensor with the slip systems. ${\dot{\gamma }}_{0}$ and m are the normalization factor and strain-rate sensitivity exponent, respectively. The rate sensitivity was assumed to be 0.05.

The threshold stress, ${\hat{\tau }}^{s}$, represents the resistance to motion of the slip systems and usually increases with deformation. The evolution of the threshold stress can be described with the Voce hardening model,
(3)$${\hat{\tau }}^{s}=\tau_{0}^{s}+\left(\tau_{1}^{s}+\theta_{1}^{s}\Gamma \right)\left(1-exp\left(-\Gamma \left|\frac{\theta_{0}^{s}}{\tau_{1}^{s}}\right|\right)\right)$$
where $\Gamma =\sum_{s}▵\gamma^{s}$ is the accumulated shear strain. $\tau_{0},\theta_{0},\theta_{1}$, and $\left(\tau_{0}+\tau_{1}\right)$ are the initial critical resolved shear stress (CRSS), initial hardening rate, asymptotic hardening rate, and the back-extrapolated CRSS, respectively. The strain hardening parameters of the Al-Si-Mg alloys are listed in Table 3.

The flow curve obtained from the uni-axial tension of the Al-Si-Mg alloys was used to determine the strain hardening parameters listed in Table 3. Only the {111}〈110〉 slip systems were assumed to operate, and the latent hardening parameter was 1.4. The experimental flow curve of the Al-Si-Mg alloys is compared with the predicted curve in Figure 2. Five hundred single crystals with a random distribution were used as initial textures to verify the VPSC modeling. This was done because the as-received TRC strips possessed an inhomogeneous texture distribution through the thickness direction, and the initial texture completely changed into a deformation texture after a 75% RA. It should be noted that the initial texture can be disregarded under large deformation.

In order to elucidate the texture evolution during the rolling process, nine texture components were selected, as listed in Table 4. They consisted of deformation textures frequently found during the PSC of the FCC metals and certain shear components. Figure 3 presents the 111 pole figures (PFs) with the ideal locations of the texture components listed in Table 4.

## 4. Results and Discussion

Shear deformation can be caused by geometry factors, defined as l/h during the rolling process. Here, *l* is the projected length of contact between the roll and the workpiece, and *h* is the mean thickness of the workpiece. The effects of geometry factors on the texture evolution were investigated, using a symmetric model domain. Figure 4 shows the variation of the geometry factors obtained from experiments and FE predictions. Overall, the geometry factor increases with an increase in the RA. It should also be noted that the number of rolling passes to produce a total RA of 75% varies depending on each rolling-pass RA. For the given RA of each rolling-pass, the first pass usually possesses the smallest geometry factor among all passes, as the diameter of the work roll was fixed, and the thickness of the strips decreased with an increase in the number of rolling passes.

The effects of the RA on the texture evolution during symmetric rolling were examined using 111 PFs in detail, as shown in Figure 5. Here, *s* is the thickness parameter, given by $s=2l/t_{0}$, where l is the distance from the center to the region of interest, and $t_{0}$ denotes the thickness. As-casted strips with a thickness of 4 mm possessed a mixed texture through the thickness direction, which mainly consisted of shear textures of 111//ND (the plane normal direction of the strips) on the surface (s = 1.0). In the center layer (s = 0), ideal rolling (or PSC) textures with some deviations from the ideal positions developed. During the rolling process, the typical deformation textures of FCC metals consist of α (Goss-Brass) and β (Brass-S-Copper) fibers as rolling textures, and some shear texture components of the rotated cube. RAs of 8.0% and 11.4% produced homogeneous rolling textures through the thickness direction. RAs between 15.5% and 36.0% revealed a strong shear texture component in the surface layer (s = 1.0). In the center layer (s = 0.0), rolling textures developed regardless of the RA.

Detailed volume fractions of each texture component listed in Table 4 are presented in Figure 6. The volume fraction of each texture component was computed using a cut-off value of 15 degrees, assuming the standard deviation of the texture components discussed in previous work [36]. The as-casted strips mainly contained shear textures of {111}〈011〉 and {111}〈112〉 in the surface layer. A minor {112}〈110〉 component also existed. Rotated cubes as a shear texture were rarely found. Figure 6a,b present the experimental results of the surface and center layers based on Figure 5. During symmetric rolling with RAs between 36% and 15.5%, rotated cubes mainly developed in the surface layer of the strips (Figure 6a). For smaller RA values of 11.4% and 8%, strong β fibers (Brass–S–Copper) developed in the surface layer. In the center layers (Figure 6b), β fibers mainly developed for all RAs, although the volume fractions of β fibers in the 36% RA case were slightly smaller compared to those for other RAs.

Figure 7 illustrates 111 PFs computed from the aforementioned symmetric model domain, as shown in Figure 1a. Figure 7a,b correspond to 111 PFs computed from E1 to E5 with a friction coefficient of 0.1. E1 and E5 correspond to the center and the surface layer of the workpiece, respectively. The evolution of the texture depends on both the location of the deformed layer and the friction coefficient. The surface layer (E5) reveals more shearing compared to the center layer (E1). With an increase in the friction coefficient, texture shearing is more evident with the rotation of the pole intensity along the transverse direction (TD). This trend is also found at an RA of 36%, as shown in Figure 8. Increased RAs usually result in more shearing in the surface layer.

Figure 9 presents the predicted volume fractions of each texture component, as computed from symmetric model domains according to various RAs and friction coefficients. As discussed in relation to the symmetric model domain, all five elements through the thickness direction were considered. Note that both the RAs and the friction coefficients affect the volume fractions of each texture component. In the surface, β fibers vary depending on both the RA and friction coefficient. The increased RAs and friction coefficients result in the decrease in the amount of β fibers. In the center, β fibers strongly developed under most RAs and friction coefficients. Under RAs of 24.5% and 36.0% at the highest friction coefficient of 0.9 (Figure 9c,d); however, fewer β fibers were predicted. This reflects the penetration of shear deformation close to the center layer under such severe rolling conditions. PSC is still valid in the center layer under symmetric rolling conditions.

Variations of the shear strain of $\varepsilon_{xy}$ during symmetric rolling with each rolling pass at an RA of 8.0% (total RA of 75%) are presented in Figure 10. All 16 steps required for the total 75% RA were examined. In the symmetric model domain, all six nodal points through the thickness direction were considered. $\varepsilon_{xy}$ values under a low friction coefficient of 0.1 are given in Figure 10a,b. With an increase in the number of rolling steps at a friction coefficient 0.1, the shear strain of $\varepsilon_{xy}$ gradually decreases. The magnitude difference of $\varepsilon_{xy}$ among the nodal points also decreases. Those under a high friction coefficient of 0.4 are presented in Figure 10c,d. The magnitudes of the shear strain $\varepsilon_{xy}$ at a friction coefficient of 0.4 are larger than those at a friction coefficient of 0.1. With an increase in the number of rolling steps, the shear strain of $\varepsilon_{xy}$ at a friction coefficient of 0.4 increases slightly. The nodal point located along the surface, Pt1, possessed greater $\varepsilon_{xy}$ values than the nodal point located along the center, Pt6. Shear texturing is expected more on the surface than in the center. In fact, shear texturing was not that excessive at a shear strain value of 0.02 according to Figure 9a.

Variations of the shear strain of $\varepsilon_{xy}$ during symmetric rolling at an RA of 36.0% are presented in Figure 11. All three steps required for the accumulated 75% RA are examined depending on various friction coefficients. With an increase in the number of rolling steps from the first to the third case, $\varepsilon_{xy}$ decreases at a friction coefficient of 0.1, as shown in Figure 11a. Increased friction coefficients of 0.4, 0.6 and 0.9 resulted in increased values of $\varepsilon_{xy}$. With an increase in the number of rolling steps, $\varepsilon_{xy}$ increases slightly. These variations are similar to those after symmetric rolling with an RA of 8.0%. The $\varepsilon_{xy}$ value when the friction coefficient is 0.9 approaches 0.4, which is much greater than that when the friction coefficient is 0.1. Much more shear deformation was produced under the former condition than the latter.

The DSR process is designed to produce shear textures effectively through the thickness direction with individual control of the top and bottom rolls. Load variations during the DSR process were examined, as shown in Figure 12. Figure 12a illustrates the experimental load variation obtained from the work rolls during the DSR of Al-Si-Mg strips. Initial strips with a thickness of 4 mm were cold-rolled down to 1 mm via five rolling passes. The average values of RA for each rolling pass were approximately 24.5%. Conventional rolling at a speed ratio of 1.00 produced the largest load on the rolls. It was found that the load decreased quickly at speed ratios between 1.0 and 1.4, after which it decreased slowly between 1.4 and 1.6. Past a speed ratio of 1.6, it quickly decreased again. The average values of the load on the rolls decreased with an increase in the differential speed ratio. This implies that the differential speed rolling process requires less work with an increase in the speed ratio compared to the conventional equal speed rolling process to produce the same thickness reduction. The predicted load variations are presented in Figure 12b for comparison. For a friction coefficient of 0.2, the loads decreased with an increase in the speed ratio between 1.0 and 1.2, after which it remained at a similar value. Under a friction coefficient of 0.4, the loads gradually decreased with an increase in the speed ratio between 1.0 and 1.4. The load changed slightly at speed ratios greater than 1.4. Based on the predicted loads on the rolls, it is evident that a greater friction coefficient increases the load on the rolls. Hence, both the speed ratio and friction coefficient affect the load on the rolls. There is some difference in the loads on the rolls between the experiments and the predictions. The experimental load on the rolls continues to decrease with an increase in the speed ratio, but the predicted value decreases and then levels out. It also appears that the friction coefficients vary with the speed ratio, and the constant values in these cases assumed during FE predictions presumably caused the difference.

Figure 13 displays 111 PFs during DSR. The total RA was 75%, and a single RA of each rolling pass was approximately 24.5%, as shown in Figure 12a. When considering the top surface layer (s = 1.0), strong shear textures of the rotated cube developed at speed ratios of 1.0 and 1.1. The top surface layer came into contact with the top roll at a constant speed. At a speed ratio of 1.2, the top surface layer revealed very strong rolling textures with minor shearing. A minor deviation from the rolling textures was confirmed approximately 10 degrees of rotation along the TD. At speed ratios of 1.3 and 1.4, strong shear textures near the rotated cube component developed from the top surface to the center. In the top middle layer (s = 0.5), a strong rotated cube was found at a speed ratio of 1.0. At speed ratios of 1.1 and 1.2, the typical rolling textures of β fibers developed. A speed ratio of 1.3 or more produced a strong rotated cube, similar to the top surface layer. In the center layer (s = 0), rolling textures developed at speed ratios of 1.0, 1.1, and 1.2. Past a speed ratio of 1.3, a strong rotated cube developed similar to the other layers. Note that strong shear texturing occurred along the thickness direction at speed ratios of 1.3 and 1.4 and greater. In addition, 111 PFs for speed ratios greater than 1.4 were very similar to those at a speed ratio of 1.4 and thus are omitted here. Inhomogeneous textures with shear and rolling components developed at speed ratios less than 1.3. Past a speed ratio of 1.3, homogeneous shear textures developed through the thickness direction.

Figure 14 illustrates the predicted 111 PFs computed from the velocity gradient obtained from FE modeling, as shown in Figure 1b. During the FE modeling process, a friction coefficient of 0.4 was applied. The first row (Figure 14a–j) displays 111 PFs corresponding to the position of E1 to E10 at a speed ratio of 1.0. The second, third, and fourth rows show 111 PFs at speed ratios of 1.1 (Figure 14k–t), 1.2 (Figure 14u–ad), and 1.4 (Figure 14ae–an), respectively. At a speed ratio of 1.0, Figure 14f,h,j correspond to the center layer (s = 0.0), the top middle layer (s = 0.5), and the top surface layer (s = 1.0), respectively. When comparing the predictions to the experiments, the dominant rolling texture in the center is similar. In the top middle and top surface regions, shear textures are found. The predicted results are close to the 〈111〉//ND shear textures, and the experimental outcomes show a rotated cube, {100}〈011〉. These different shear texturing behaviors call for further studies. Under a symmetric rolling condition at a speed ratio of 1.0, the center layer of the sheets usually possessed a rolling texture, and both surface layers revealed shear textures. The shear texture distributions for both surface layers show a relationship of a 180 degree rotation along the TD. At a speed ratio of 1.2, experimental deformation textures obtained from the center to the top surface layers, as shown in Figure 13d,j,p, were rolling textures. This trend was also found in the predicted pole figures corresponding to the center to the top surface layer, as shown in Figure 14z,ab,ad. Deformation textures in the bottom roll side continued to possess shear textures. Note that shear texturing along the thickness direction was enabled under DSR with a speed ratio greater than approximately 1.2.

More detailed texture evolution during the rolling process was examined based on the volume fraction of each texture component. From the volume fraction of each texture component obtained from the top surface, dynamic variation of the texture volume fraction with the speed ratio was evident (Figure 15a). The rotated cube component increased drastically, even during the symmetric rolling process at a speed ratio of 1.0, although there was a slightly rotated cube initially. With an increase in the speed ratio to 1.20, the rotated cube component gradually decreased. Instead, β fibers (Brass–S–Copper) sharply increased in number. At a speed ratio of 1.20, the greatest volume fraction of β fibers was observed, while the rotated cube was negligible. At a speed ratio of 1.25, the β fibers were still dominant, and the rotated cube increased slightly. At a speed ratio of 1.30 or more, the rotated cube component was dominant. Figure 15b reveals the volume fraction of the texture components in the center layer. As discussed in relation to Figure 13, initial strips contained α and β fibers. During the rolling process, β fibers increased drastically, and the Goss texture component nearly disappeared. These trends appeared at speed ratios of up to 1.25. At a speed ratio of 1.30 or more, a strong rotated cube developed.

Based on the predicted texture evolution obtained from the DSR domain, the volume fraction of each texture component along the thickness direction is shown in Figure 15c. Variation of the volume fraction along 10 elements (E1–E10), as shown in Figure 1b), is evident. The workpieces in contact with the bottom and top rolls correspond to elements E1 and E10, respectively. At a speed ratio of 1.0, the volume fraction of the texture components appears to be symmetric due to the conventional symmetric rolling condition. The center elements (E5 and E6) illustrate the dominant volume fractions of β fibers. Near the surface of the workpiece, the rolling textures decrease, and shear textures increase. At a speed ratio of 1.10, E1 near the bottom roll possesses increased shear textures, and E10 near the top roll shows strong rolling textures. With an increase in the speed ratio, shear textures increase along the thickness direction, ranging from E1 to E10. It is predicted that a certain number of rolling textures will remain, even at a high differential speed ratio. Experimental results, however, reveal a drastic decrease in rolling textures. Severe shear texturing during DSR can be caused by high friction, which could vary during the DSR process. The effects of varying friction coefficients on texturing will be left as future work.

Figure 16 presents the shear strain distribution during DSR modeling with a friction coefficient of 0.4. Each rolling pass utilized an RA of approximately 24.5%, and the overall RA was 75% with five rolling passes. A DSR simulation was also conducted through five steps. At a speed ratio of 1.0, the shear strain of $\varepsilon_{xy}$ reveals a symmetric distribution along the 11 nodal points or tracking points. Each nodal point is specified in Figure 1b. The shear strain of $\varepsilon_{xy}$ increases slightly with an increase in the number of rolling steps, similar to the symmetric condition, as discussed in Figure 11b. At a speed ratio of 1.1 or more, all shear strains of $\varepsilon_{xy}$ along each nodal point shift to negative values. At speed ratios of 1.0 and 1.1, the magnitude of the shear strain $\varepsilon_{xy}$ along each nodal point reveals a sequential increase with the number of steps. However, at speed ratios of 1.2 and 1.4, $\varepsilon_{xy}$ decreases slightly or remains similar to the increased numbers of steps. At a speed ratio of 1.4, all shear strains of $\varepsilon_{xy}$ along each nodal point converge tightly to similar values, implying that similar shear texturing occurs along all nodal points. A large value of the shear strain $\varepsilon_{xy}$ also resulted in strong shear texturing along the thickness direction.

## 5. Conclusions

Two different rolling processes, symmetric rolling with a different reduction in area (SRDRA) and differential speed rolling (DSR), were examined in order to gain a detailed understanding of the crystallographic texture evolution and shear strain distributions during the cold rolling of Al-Si-Mg (1.0%Si-0.6%Mg) aluminum strips.

Initial Al-Si-Mg (1.0%Si-0.6%Mg) aluminum strips fabricated by twin-roll casting possessed inhomogeneous texture distributions along the thickness direction, i.e., shear textures of 〈111〉//ND fibers in the surface layer and rolling or PSC textures (α fiber, Goss–Brass and β fiber, Brass–S–Copper components) in the center layer.During the SRDRA process with a total RA of approximately 75%, the shear texture of a rotated cube, {100}〈011〉, mainly occurred at the surface layer, and typical rolling textures of β fibers were observed at the center layer. At RAs of 7.8% and 11.4% for each rolling pass, typical rolling textures developed at all thicknesses. Increased RAs for each rolling pass greater than approximately 15% resulted in shear textures at the surface and middle layers. A rolling texture occurred in the center, regardless of the RA. Using a symmetric FE model domain along with a viscoplastic constitutive equation for Al-Mg-Si alloys, the texture evolution was predicted with various RAs and friction coefficients. With an increase in the friction coefficient, shear textures clearly increased. Distribution of the shear strain $\varepsilon_{xy}$ along the tracking points through the thickness direction reflected shear texture evolution.The DSR process with a total RA of 75% (each rolling pass having a 25% RA) caused more dynamic texture evolution than SRDRA. Both shear and PSC textures along the thickness direction varied with the speed ratio. The top surface layer in contact with the roll operating at a lower speed revealed dynamic variations of the rotated cube and β fibers. At speed ratios lower than 1.2, a strong rotated cube developed. At roll speed ratios near 1.2, β fibers developed. At roll speed ratios greater than 1.2, a rotated cube developed again. Shear texturing along the thickness direction was enabled under DSR with speed ratios greater than approximately 1.3. Based on FE modeling covering the entire thickness range, shear deformation was predicted. The predicted textures in the top surface layer were similar to the experimental textures. The bottom surface layer in contact with the roll running at a higher speed continued to possess shear textures regardless of the speed ratio. The shear strain, $\varepsilon_{xy}$, along the tracking nodal points was closely associated with the shear texture evolution. Material points with greater $\varepsilon_{xy}$ values than 0.1 resulted in strong shear texturing.During the DSR process, the load in rolls decreases with the speed ratio. An increase in the friction coefficient resulted in an increase in the load in rolls. The geometry factor to be defined by the projected area of the roll onto the sheet increases with an increase in the RA. Shear texturing usually increases with the geometry factor.

## Figures and Tables

**Figure 1 materials-17-00179-f001:**
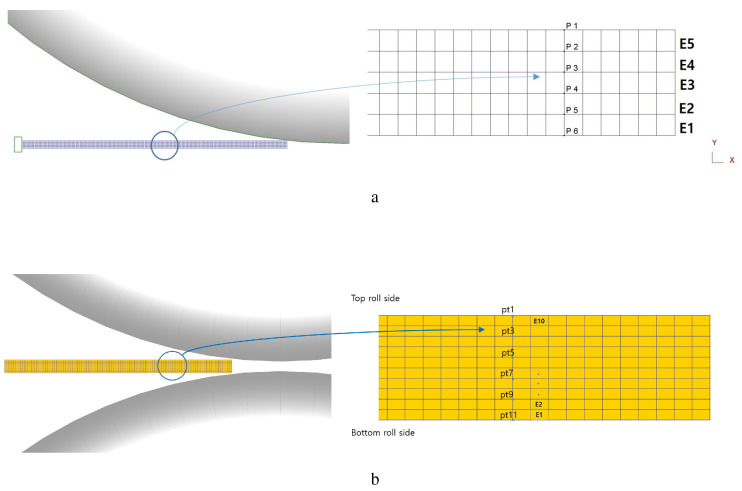
Schematic diagram to illustrate the (**a**) symmetric and (**b**) asymmetric model domains of the rolling processes.

**Figure 2 materials-17-00179-f002:**
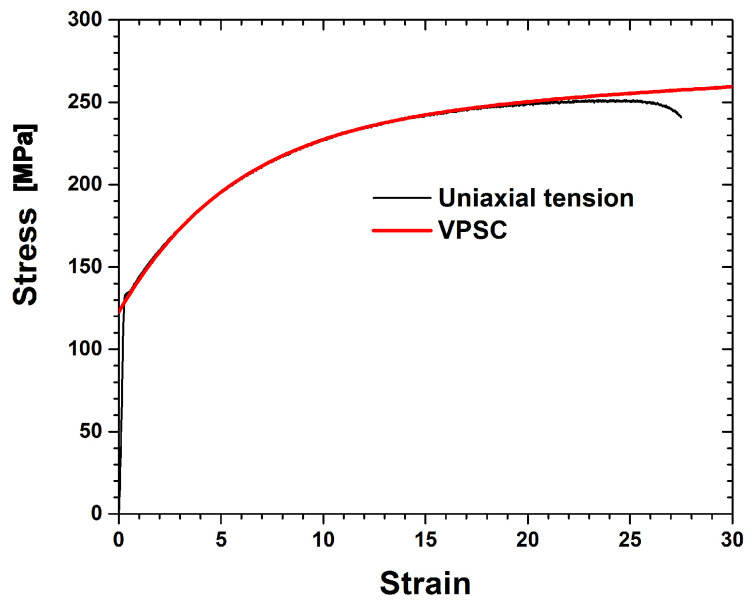
Flow curves obtained from the experiment and VPSC model prediction of Al-Si-Mg alloys.

**Figure 3 materials-17-00179-f003:**
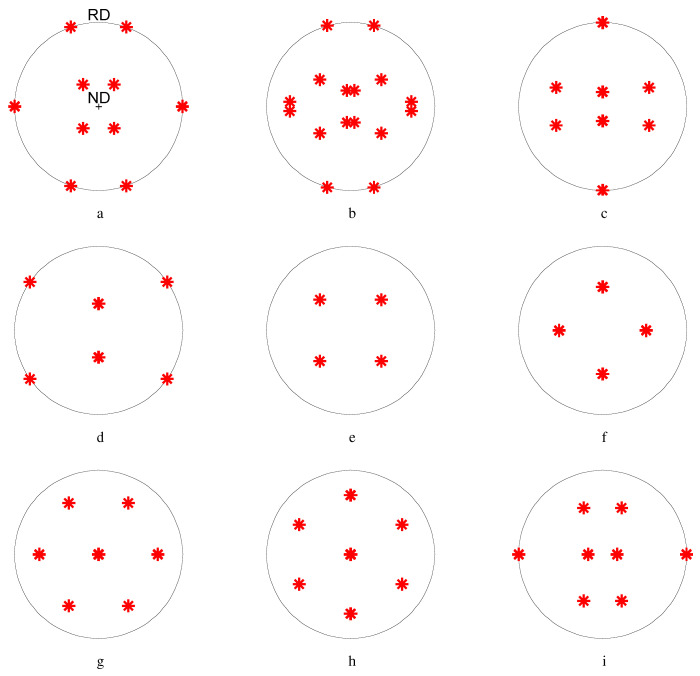
Standard texture components displayed in 111 pole figures (PFs), as listed in Table 4. RD: the rolling direction; ND: the normal direction. (**a**) Brass, (**b**) S, (**c**) copper, (**d**) goss, (**e**) cube, (**f**) rotated cube, (**g**) E, (**h**) F, and (**i**) Y.

**Figure 4 materials-17-00179-f004:**
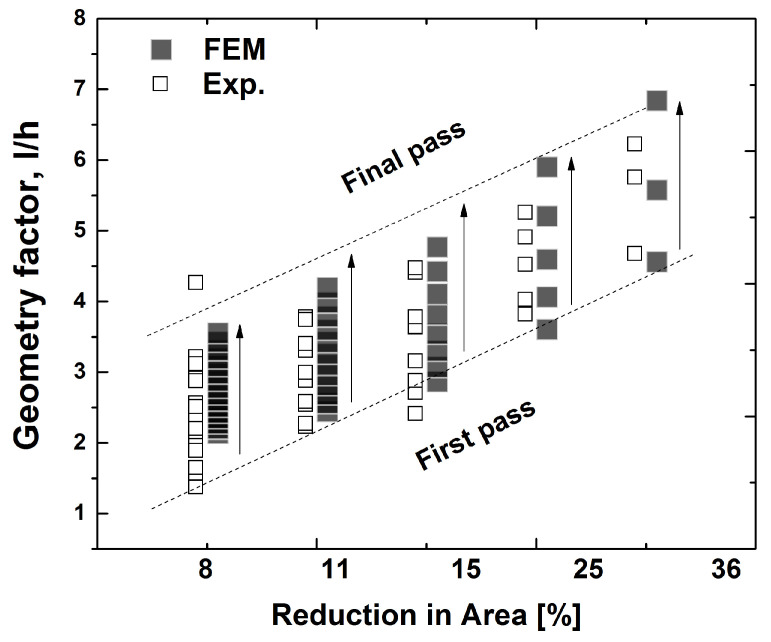
Variation of the geometry factor with a reduction in area. As shown in Table 1, 8% of RA possessed 17 rolling passes, and thus 17 gray grids are overlapped.

**Figure 5 materials-17-00179-f005:**
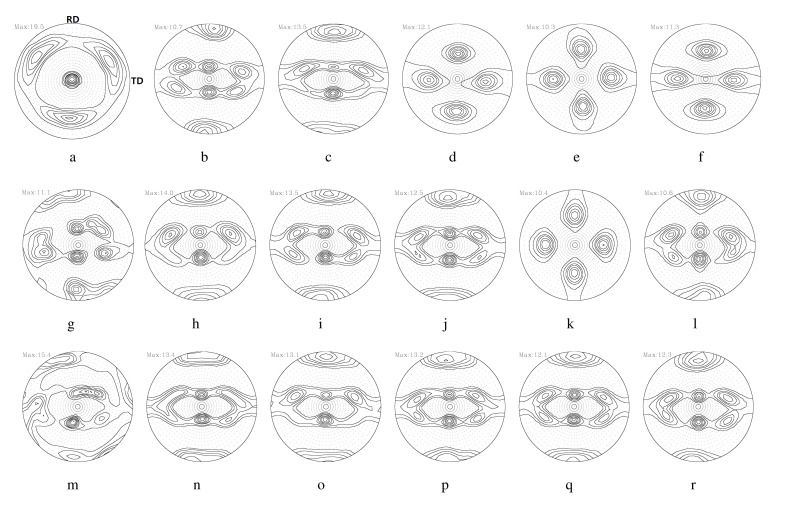
Experimental 111 PFs of Al-Mg-Si strips when rolling with various RAs. Surface layers (s = 1.0): (**a**) as-cast and with RAs of (**b**) 8.0%, (**c**) 11.4%, (**d**) 15.5%, (**e**) 24.5%, and (**f**) 36.0%. Middle layers (s = 0.5): (**g**) as-cast and with RAs of (**h**) 8.0%, (**i**) 11.4%, (**j**) 15.5%, (**k**) 24.5%, and (**l**) 36.0%. Center layers (s = 0.0): (**m**) as-cast and with RAs of (**n**) 8.0%, (**o**) 11.4%, (**p**) 15.5%, (**q**) 24.5%, and (**r**) 36.0%.

**Figure 6 materials-17-00179-f006:**
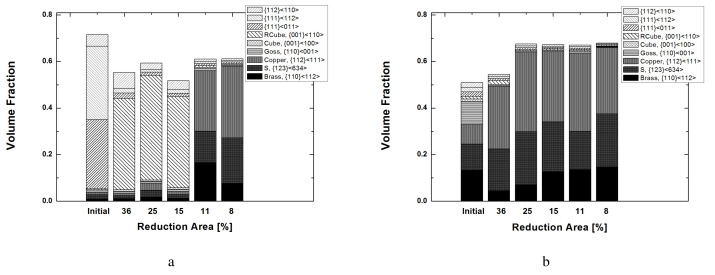
Experimental volume fractions of each texture component of Al-Mg-Si strips when rolled with various RAs: (**a**) surface and (**b**) center layers.

**Figure 7 materials-17-00179-f007:**
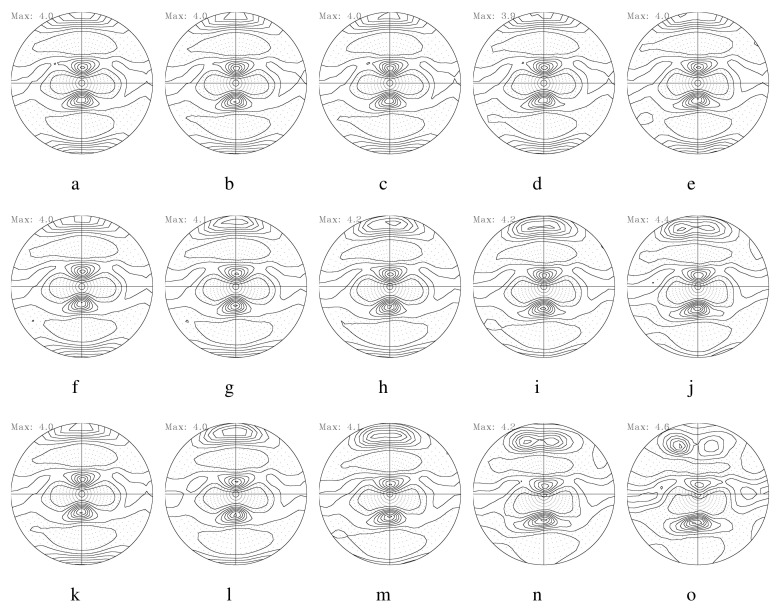
Predicted 111 PFs of Al-Mg-Si strips during rolling with an RA of 8% per rolling pass. The first row displays 111 PFs from the center (**a**) to the surface (**e**) with a friction coefficient of 0.1. (**b**–**d**) are PFs between the center and the surface. The second (**f**–**j**) and third (**k**–**o**) rows correspond to those with friction coefficients of 0.4 and 0.6, respectively.

**Figure 8 materials-17-00179-f008:**
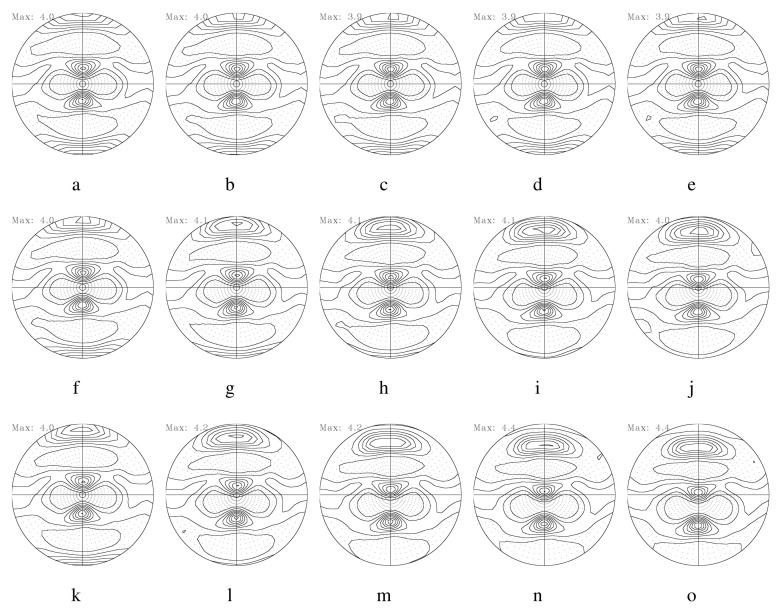
Predicted 111 PFs of Al-Mg-Si strips during rolling with an RA of 36% per rolling pass. The first row displays 111 PFs from the center (**a**) to the surface (**e**) under a friction coefficient of 0.1. (**b**–**d**) are PFs between the center and the surface. The second (**f**–**j**) and third (**k**–**o**) rows correspond to those under friction coefficients of 0.4 and 0.6, respectively.

**Figure 9 materials-17-00179-f009:**
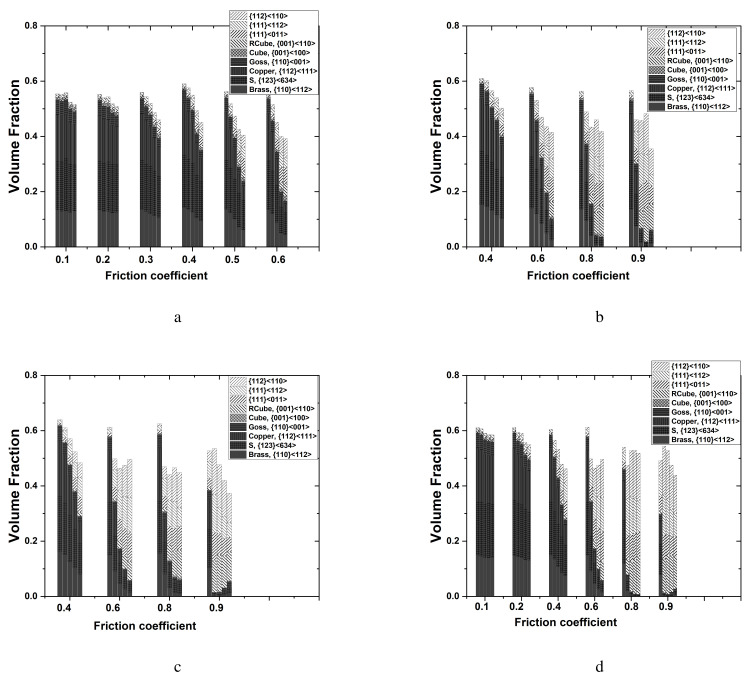
Predicted volume fractions of each texture component of Al-Mg-Si strips during rolling with various RAs. Predicted results with different friction coefficients are given with RAs of (**a**) 8.0%, (**b**) 15.5%, (**c**) 24.5%, and (**d**) 36.0%.

**Figure 10 materials-17-00179-f010:**
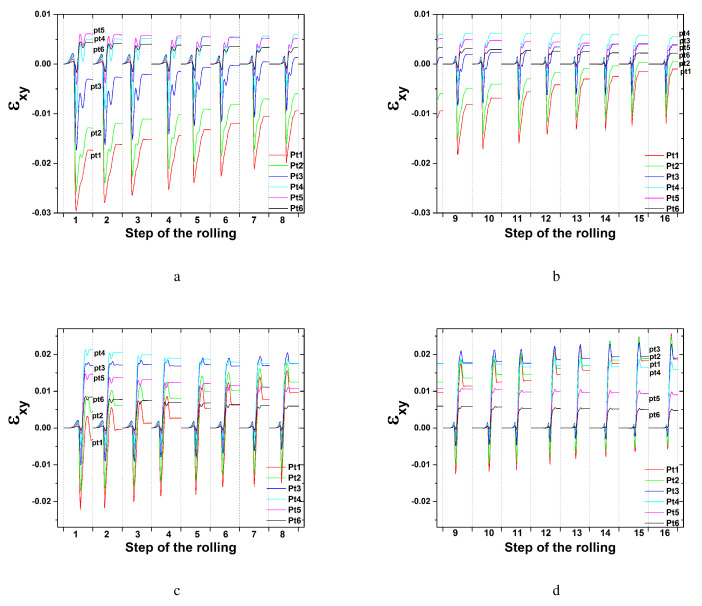
Shear strain distribution during symmetric rolling with an RA of 8.0% at various friction coefficients: (**a**) the first to eighth step and (**b**) the ninth to sixteenth step at a friction coefficient of 0.1, and (**c**) the first to eighth step and (**d**) the ninth to sixteenth step at a friction coefficient of 0.4.

**Figure 11 materials-17-00179-f011:**
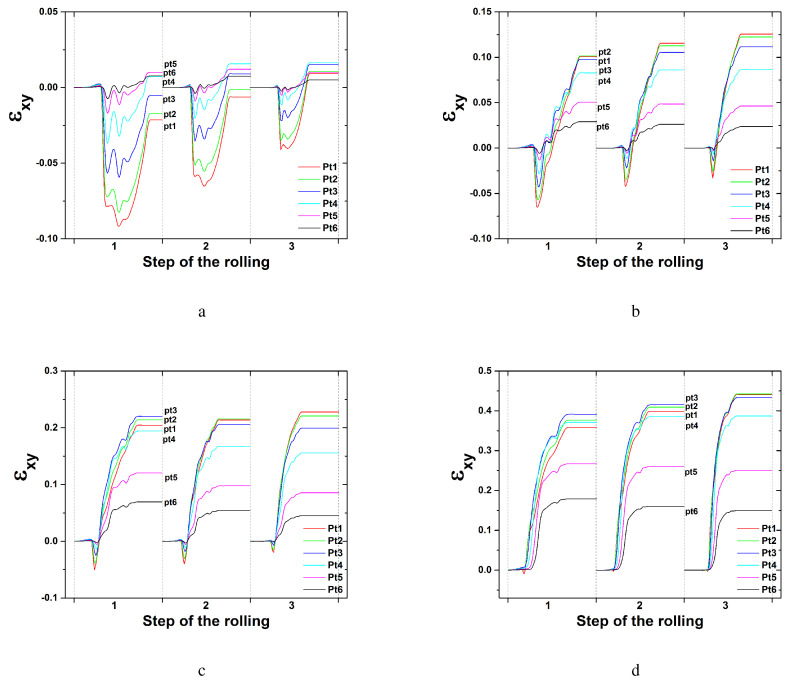
Shear strain distribution during symmetric rolling with an RA of 36.0% at various friction coefficients: (**a**) 0.1, (**b**) 0.4, (**c**) 0.6, and (**d**) 0.9.

**Figure 12 materials-17-00179-f012:**
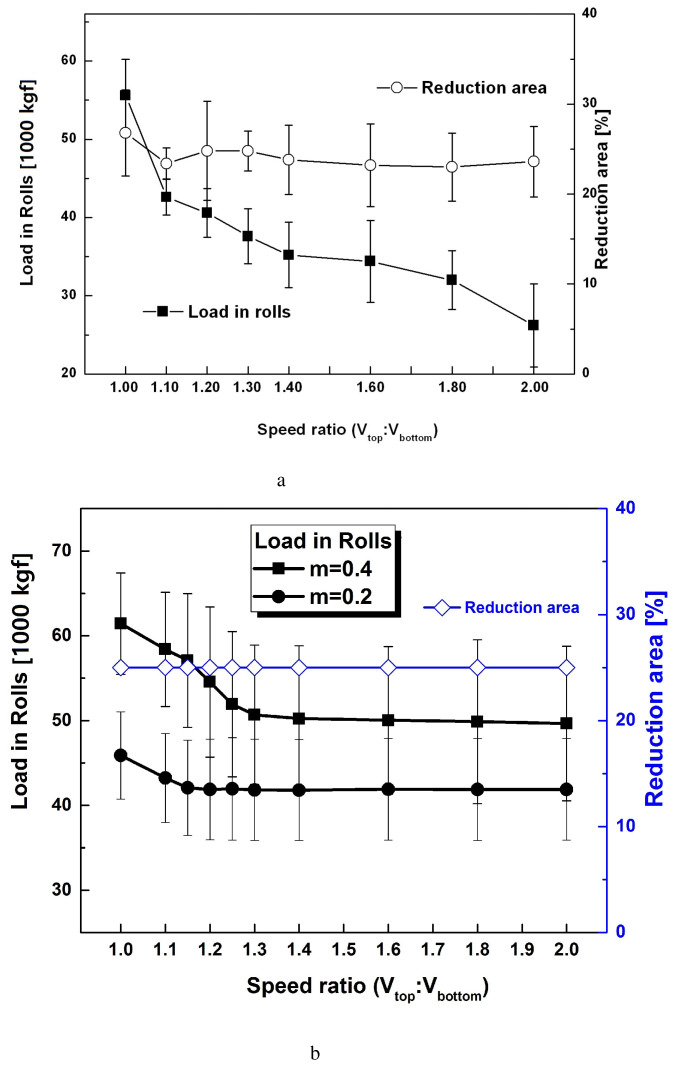
Load variation during differential speed rolling of Al-Mg-Si strips: (**a**) experiments and (**b**) predictions.

**Figure 13 materials-17-00179-f013:**
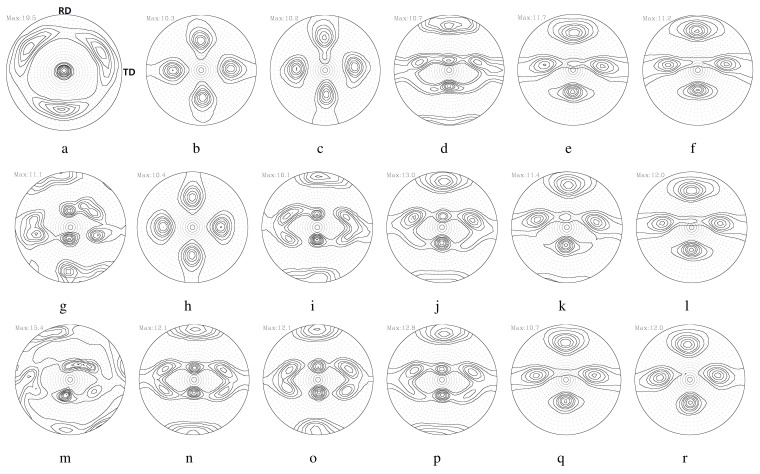
Experimental 111 PFs during the DSR of Al-Mg-Si strips. Top surface layers (s = 1.0): (**a**) as-cast and at speed ratios of (**b**) 1.0, (**c**) 1.1, (**d**) 1.2, (**e**) 1.3, and (**f**) 1.4. Top middle layers (s = 0.5): (**g**) as-cast and at speed ratios of (**h**) 1.0, (**i**) 1.1, (**j**) 1.2, (**k**) 1.3, and (**l**) 1.4. Center layers (s = 0.0): (**m**) as-cast and at speed ratios of (**n**) 1.0, (**o**) 1.1, (**p**) 1.2, (**q**) 1.3, and (**r**) 1.4.

**Figure 14 materials-17-00179-f014:**
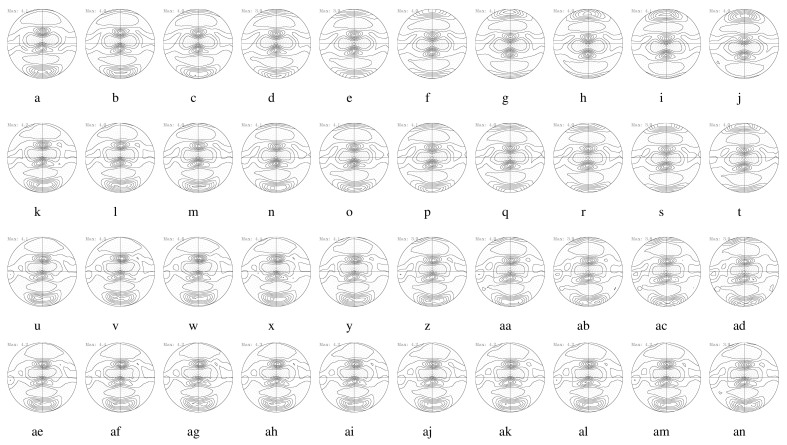
Predicted 111 PFs during the DSR of Al-Mg-Si strips. Ten PFs along each row were obtained from 10 elements along the thickness direction in the DRS model domain. The first row displays 111 PFs at a speed ratio of 1.0 (**a**–**j**). The 111 PFs located along the second (**k**–**t**), third (**u**–**ad**), and fourth (**ae**–**an**) rows correspond to speed ratios of 1.1, 1.2, and 1.4, respectively.

**Figure 15 materials-17-00179-f015:**
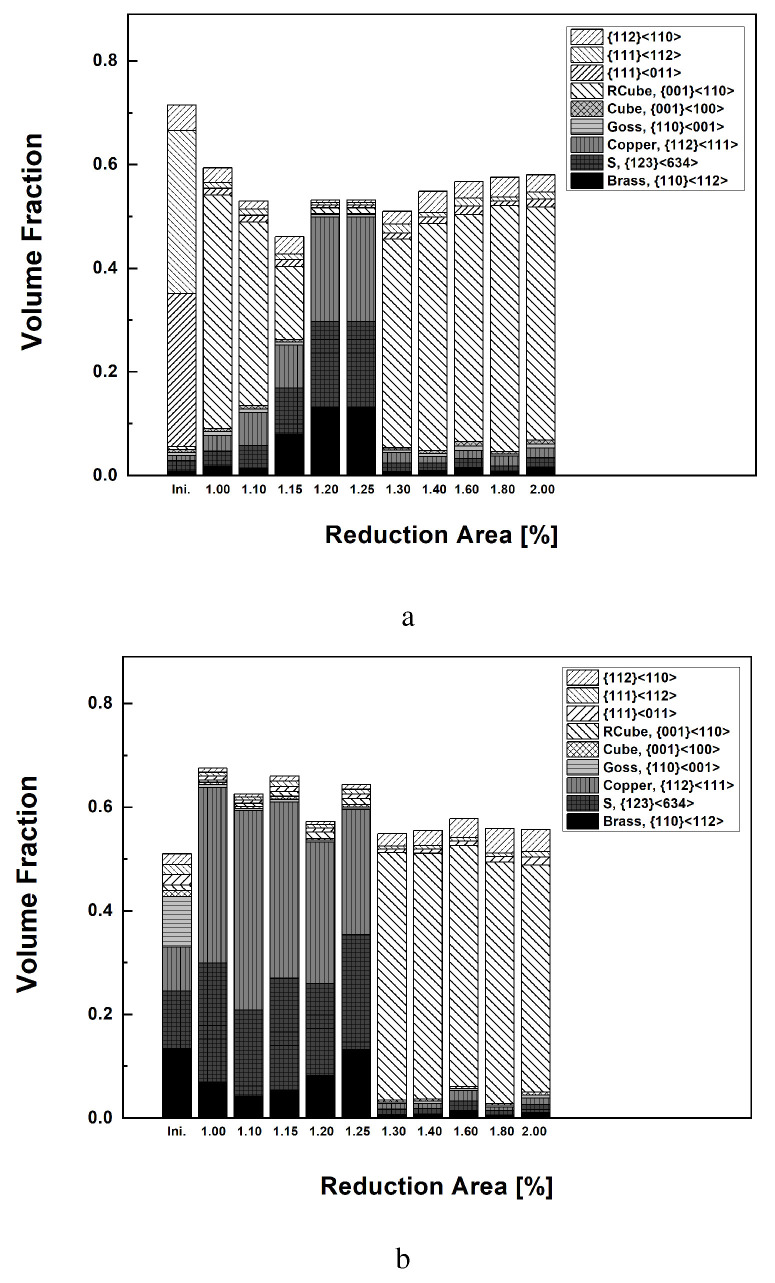
Volume fraction of texture components of Al-Mg-Si strips during DSR. Experimental results: (**a**) the top surface and (**b**) the center. Predicted results: (**c**) the whole thickness (E1–E10, as shown in Figure 1b).

**Figure 16 materials-17-00179-f016:**
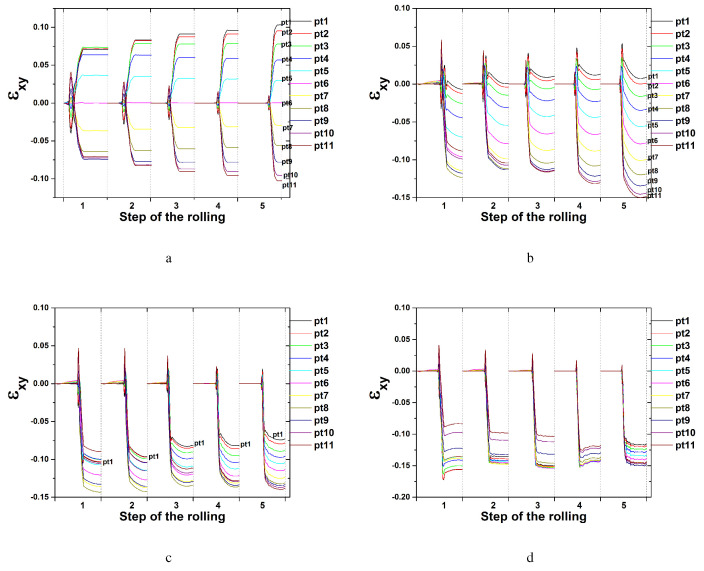
Shear strain distribution during DSR at various speed ratios: (**a**) 1.0, (**b**) 1.10, (**c**) 1.20, and (**d**) 1.40.

**Table 1 materials-17-00179-t001:** Various working parameters for symmetric rolling.

Speeds	RA per Pass	Total Rolling Passes	Total RA
[mpm]	[%]	[Nr.]	[%]
5	8.0	17	75
5	11.4	11	75
5	15.5	9	75
5	24.5	5	75
5	36.0	3	75

*Note:* mpm: meters per minute; RA: reduction in area.

**Table 2 materials-17-00179-t002:** Various working parameters for differential speed rolling.

Working Parameters	Values
RA per pass [%]	24.5
Total RA [%]	75
Constant top roll speed [mpm]	5
Various bottom roll speed [mpm]	5.0, 5.5, 6.0, 6.5, 7.0, 8.0, 9.0, 10.0
Differential speed ratio	1.0, 1.1, 1.2, 1.3, 1.4, 1.6, 1.8, 2.0

*Note:* mpm: meters per minute; RA: reduction in area.

**Table 3 materials-17-00179-t003:** Strain hardening parameters for a viscoplastic self-consistent (VPSC) model for Al-Si-Mg alloys.

Slip Systems {hkl}〈uvw〉	$\tau_{0}$ [MPa]	$\tau_{1}$ [MPa]	$\theta_{0}$ [MPa]	$\theta_{1}$ [MPa]
{111}〈110〉	62	62	455	8

**Table 4 materials-17-00179-t004:** Standard orientation components.

Designation	Miller Indices {hkl}〈uvw〉	Euler Angles $\left\{\phi_{1},\Psi ,\phi_{2}\right\}$
Brass	{110}〈112〉	{35.264590}
S	{123}〈634〉	{58.9836.763.43}
Copper	{112}〈111〉	{9035.2645}
Goss	{110}〈001〉	{909045}
Cube	{001}〈100〉	{000}
Rcube	{001}〈110〉	{4500}
E	{111}〈110〉	{6054.74315}
F	{111}〈112〉	{9054.74315}
Y	{112}〈110〉	{18035.26315}

## Data Availability

Data are contained within the article.

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
