# Peer review of "Dynamic Shear Texture Evolution during the Symmetric and Differential Speed Rolling of Al-Si-Mg Alloys Fabricated by Twin Roll Casting"

_materials, 2023, doi:10.3390/ma17010179_

Round 1
Reviewer 1 Report
Comments and Suggestions for Authors
The manuscript is on evolution of deformation texture of Al-Mg-Si alloy during symmetric and differential speed rolling. The manuscript is well formulated, and results are interesting. However, there are issues with the quality of certain figures. Therefore, requested to improve resolution of the figures in the manuscript. Also, no microstructures were presented for aluminum strips fabricated by twin-roll cast. Starting microstructure has huge role in deformation texture. Was the microstructure homogeneous? How was the grain structure?
Author Response
Thank you for the valuable comments. We revised as below according to the reviewer's comments.
The manuscript is on evolution of deformation texture of Al-Mg-Si alloy during symmetric and differential speed rolling. The manuscript is well formulated, and results are interesting.
However, there are issues with the quality of certain figures. Therefore, requested to improve resolution of the figures in the manuscript.
->We replaced old figures with higher resolution images.
Also, no microstructures were presented for aluminum strips fabricated by twin-roll cast. Starting microstructure has huge role in deformation texture. Was the microstructure homogeneous? How was the grain structure?
-> Here are some information (EBSD data) about initial microstructure of twin-roll casted strip 6 mm thick. However, we are sorry that we don’t have the matched grain structure of twin-roll casted strip 4 mm thick. We think overall grain structure would be similar for those TRC strips.

Reviewer 2 Report
Comments and Suggestions for Authors
In this study, texture evaluation of al-si-mg alloys produced by twin rolling casting method was made depending on different rolling speeds. The article is complex in organization and presentation. Additionally, when the references were examined, very old sources were used. Generally , it is not appropriate to accept it.
No information about rolling conditions is given in the summary section.
Although the subject is explained in the introduction, previous studies in the literature have not been sufficiently examined and their differences with this subject have not been revealed.
The chemical composition of the material is not given.
Test conditions should be given in a table (speed etc.).
The article only contains information in terms of texture. For this reason, in my opinion, the publication is not sufficient in terms of depth.
Figure 9 is added as a screenshot and there are elements on the figure that are not related to the graphic.
Figure 9 is given in sufficient gray scale and is very difficult to understand.
The references are very old.
Author Response
Thank your for your comments about this draft. We responds to your valuable comments below:
No information about rolling conditions is given in the summary section.
->Detailed information about rolling conditions, including roll diameter of 280 mm, no lubrication, total reduction in area of 75%, step reduction in areas per pass of 8%, 11.4%, 15.5%, 24.5%, and 36% are provided in the experimental section. Roll speeds vary from 5 mpm (0.595 rad/sec) to 10 mpm (1.19 rad/sec). Summarized working conditions are provided in Tables 1 and 2 on page 3
Although the subject is explained in the introduction, previous studies in the literature have not been sufficiently examined and their differences with this subject have not been revealed.
-> Most discussion on differential speed rolling focused on conventional sheets not twin-roll casted strips. Here we examined effects of various rolling parameters on texture evolution during differential speed rolling of Al-1.0Si-0.6Mg TRC strips.
-> lines 71-72 are added
The chemical composition of the material is not given.
->Chemical composition is given as Al-1.0Si-0.6Mg in the experimental section.
->Lines 87-88 are specified
Test conditions should be given in a table (speed etc.).
->We summarized the rolling conditions of symmetric and asymmetric rolling in Table 1 and Table 2, respectively.
The article only contains information in terms of texture. For this reason, in my opinion, the publication is not sufficient in terms of depth.
->We also examined microstructure of twin-roll casted strips. Here are some information (EBSD data) about initial microstructure of twin-roll casted strip 6 mm thick. However, we are sorry that we don’t have the matched grain structure of twin-roll casted strip 4 mm thick. We think overall grain structure would be similar for those TRC strips.
->Anisotropy based on texture is most important in Al sheets for automobile applications (outer panels). That’s why this paper discussed texture evolution based crystal plasticity in detail.
Figure 9 is added as a screenshot and there are elements on the figure that are not related to the graphic. Figure 9 is given in sufficient gray scale and is very difficult to understand.
->Nine texture components in Fig. 9 are typical rolling textures of fcc metals. We replaced old figure with higher resolution images for clarity
The references are very old.
->We are sorry about that. Quantitative texture analysis on metallic sheets has a long history and thus some references are old. We included more recent references.
->Lines 39-43, 55-60 new references are added

Reviewer 3 Report
Comments and Suggestions for Authors
The effects of reduction in area (RA) and speed ratio between the top and bottom rolls on shear strain and the crystallographic texture evolution of Al-Si-Mg (1.0%Si-0.6%Mg) aluminum alloys fabricated by twin roll casting (TRC) were comprehensively examined experimentally and through numerical predictions.
I consider the current work is finalized in terms of simulation, experimentation, and discussion, and I recommend its publication with minor revisions.
1. In the Experimental Procedure section, in the sentence” The obtained data were analyzed using the WIMV method with automated conditional ghost correction...” it would be helpful to define the acronym WIMV.
2. In the Simulation Procedure section, the text discusses the reasons for varying the number of elements per material section depending on the process. However, it would be beneficial to also include a discussion on the size of each element and the selected time step, considering that accurate results from simulations of severe plastic deformation processes require precise control of these parameters. The addition of these details would enhance the comprehensiveness of this section.
3. On page 18, the sentence “At a speed ratio of 1.2, experimental deformation textures obtained from the center to the top surface layers, as shown in Figs. 13d), 13j), and 13p) were rolling textures” is cut off after “top surface layers”.
4. Regarding the document format, some of the Figures are referenced several pages ahead of their appearance. For instance, Figures 9, 12, and 13 are referenced on pages 9 and 12, but they do not appear until pages 13, 16, and 17 respectively. This can disrupt the reading experience, so I suggest revising the document to ensure that Figures are referenced on the same page or on the following page.
5. It has been noted that figures 3, 9, and 12 contain an erroneous box at the bottom which should be removed. Additionally, in figures 5, 7, 8, and 13, it is recommended that the label denoting the maximum value in the upper left corner of each image be enlarged or highlighted.
6. Lastly, it is suggested that the bibliography include the title of each referenced paper, as per the template format, for ease of use in common databases.
Author Response
The effects of reduction in area (RA) and speed ratio between the top and bottom rolls on shear strain and the crystallographic texture evolution of Al-Si-Mg (1.0%Si-0.6%Mg) aluminum alloys fabricated by twin roll casting (TRC) were comprehensively examined experimentally and through numerical predictions.
I consider the current work is finalized in terms of simulation, experimentation, and discussion, and I recommend its publication with minor revisions.
- In the Experimental Procedure section, in the sentence” The obtained data were analyzed using the WIMV method with automated conditional ghost correction...” it would be helpful to define the acronym WIMV.
-> WIMV is a acronym of William-Imhof-Matthies-Vinel (WIMV), which is a methodology of quantitative texture analysis .
-> lines 107-109 were modified
- In the Simulation Procedure section, the text discusses the reasons for varying the number of elements per material section depending on the process. However, it would be beneficial to also include a discussion on the size of each element and the selected time step, considering that accurate results from simulations of severe plastic deformation processes require precise control of these parameters. The addition of these details would enhance the -comprehensiveness of this section.
->The workpieces passed the roll gap less than 0.2 sec. The thickness of the workpieces was 4 mm. For differential speed rolling, the whole domain was modeled. Each element cover about 0.4mm thickness of the strips.
-> we mentioned that information in the draft, lines 121-123
- On page 18, the sentence “At a speed ratio of 1.2, experimental deformation textures obtained from the center to the top surface layers, as shown in Figs. 13d), 13j), and 13p) were rolling textures” is cut off after “top surface layers”.
-> Thanks for the notice. We fixed it.
- Regarding the document format, some of the Figures are referenced several pages ahead of their appearance. For instance, Figures 9, 12, and 13 are referenced on pages 9 and 12, but they do not appear until pages 13, 16, and 17 respectively. This can disrupt the reading experience, so I suggest revising the document to ensure that Figures are referenced on the same page or on the following page.
->Thanks for the suggestion. We try to fit the Figures and texts on the same page or on the following page.
- It has been noted that figures 3, 9, and 12 contain an erroneous box at the bottom which should be removed. Additionally, in figures 5, 7, 8, and 13, it is recommended that the label denoting the maximum value in the upper left corner of each image be enlarged or highlighted.
-Regarding Figs. 3, 9, and 12, we removed them.
-Regarding pole figures appearing in Figs. 5, 7, 8, and 13, we replaced them with higher resolution images.
- Lastly, it is suggested that the bibliography include the title of each referenced paper, as per the template format, for ease of use in common databases.
-> We revised references with titles.

Round 2
Reviewer 2 Report
Comments and Suggestions for Authors
The publication has improved compared to its previous form. The editorial office may accept it if it deems it appropriate.
Author Response
-In this revised paper, Figure 6, Figure 9, Figure 12b, Figure 15a and b are all revised according to reviewers’ comments. The inset images are put in the frame now. The scale in the top and right coordinates are also deleted.
-More citations including experimental researches to support their simulation results were added in reference 20 and 21.
